# Small-Extracellular-Vesicle-Derived miRNA Profile Identifies miR-483-3p and miR-326 as Regulators in the Pathogenesis of Antiphospholipid Syndrome (APS)

**DOI:** 10.3390/ijms241411607

**Published:** 2023-07-18

**Authors:** Cristina Solé, Maria Royo, Sebastian Sandoval, Teresa Moliné, Josefina Cortés-Hernández

**Affiliations:** 1Rheumatology Research Group—Lupus Unit, Vall d’Hebrón University Hospital, Vall d’Hebrón Research Institute (VHIR), Universitat Autònoma de Barcelona (UAB), 08193 Barcelona, Spain; maria.royo@vhir.org (M.R.); sebastian.sandoval@vallhebron.cat (S.S.); fina.cortes@vhir.org (J.C.-H.); 2Department of Pathology, Vall d’Hebrón University Hospital, 08035 Barcelona, Spain; teresa.moline@vhir.org

**Keywords:** small extracellular vesicle, microRNA profiling, antiphospholipid syndrome, pathogenesis

## Abstract

Primary antiphospholipid syndrome (PAPS) is a systemic autoimmune disease associated with recurrent thrombosis and/or obstetric morbidity with persistent antiphospholipid antibodies (aPL). Although these antibodies drive endothelial injury and thrombophilia, the underlying molecular mechanism is still unclear. Small extracellular vesicles (sEVs) contain miRNAs, key players in intercellular communication. To date, the effects of miRNA-derived sEVs in PAPS are not well understood. We characterised the quantity, cellular origin and miRNA profile of sEVs isolated from thrombotic APS patients (PAPS, *n* = 50), aPL-carrier patients (aPL, *n* = 30) and healthy donors (HD, *n* = 30). We found higher circulating sEVs mainly of activated platelet origin in PAPS and aPL patients compared to HD, that were highly engulfed by HUVECs and monocyte. Through miRNA-sequencing analysis, we identified miR-483-3p to be differentially upregulated in sEVs from patients with PAPS and aPL, and miR-326 to be downregulated only in PAPS sEVs. In vitro studies showed that miR-483-3p overexpression in endothelial cells induced an upregulation of the PI3K-AKT pathway that led to endothelial proliferation/dysfunction. MiR-326 downregulation induced NOTCH pathway activation in monocytes with the upregulation of NFKB1, tissue factor and cytokine production. These results provide evidence that miRNA-derived sEVs contribute to APS pathogenesis by producing endothelial cell proliferation, monocyte activation and adhesion/procoagulant factors.

## 1. Introduction

Antiphospholipid-antibody syndrome (APS) is an acquired thrombophilic disorder in which vascular thrombosis (venous or arterial) and/or pregnancy losses may occur in the presence of persistent antiphospholipid antibodies (aPL) [1,2,3]. While in vitro studies and animal models have provided insight into some aspects of APS pathogenesis, mechanisms behind the syndrome are complex and not fully understood [4]. Antiphospholipid antibodies exert prothrombotic effects by interacting with coagulatory proteins and inhibitors, and by the activation of vascular cells in a β2-GPI-dependent manner [5]. As a result of this activation, there is an increased expression of proinflammatory cytokines that in turn contribute to the prothrombotic phenotype [6] and the release of extracellular vesicles (EVs) [7].

Small EVs are membrane-enclosed particles released by cells (<200 nm) as a response to various physiological processes including apoptosis, senescence, and cellular activation that play an important role in intercellular communication and represent a potential source for disease biomarkers [8,9,10,11,12]. Elevated levels of circulating medium/large EVs have been described in APS with endothelial origin [13,14,15,16,17,18,19,20,21,22] and one study has evaluated the role of small EVs (sEVs) in APS pathogenesis [20]. They showed that sEVs from APS patients contribute to endothelial and platelet activation as measured by enrichment of surface CD62P expression [20]. sEVs are attractive as diagnostic and therapeutic markers since they are relatively enriched in miRNA with unique expression profiles, participate in intercellular communication over both short and long distances, have longer lifespan, and are more resistant to protease degradation [23]. Like other autoimmune disorders, several lines of evidence support the idea that miRNAs are involved in the pathogenesis of APS, interacting with innate and adaptive immune response functions [24,25,26,27]. These studies have been able to identify distinct miRNA expression profiles in APS linked to pro-oxidative state, TF modulation and mitochondrial dysfunction, leading to inflammation and progression of atherosclerosis [25,26,27,28].

As little is known about the pathogenic role played by sEVs and their derived miRNAs in APS, we performed miRNA-seq analysis using sEVs from APS patients, patients with aPL without thrombotic events, and healthy donors (HD) to identify a differential sEV-derived miRNA profile. Further in vitro studies were conducted to understand their role in APS pathogenesis.

## 2. Results

### 2.1. Clinical Characteristics

A total of 80 patients with aPLs and 30 healthy controls were included in the study. Of the patients with aPL, 50 had thrombotic PAPS (of whom, 11 had obstetric complications), and 30 had aPL without associated complications. Patient demographics of the three groups are shown in Table 1. The majority of patients had triple antibody positivity (80%). Venous thrombosis was the most frequent thrombotic event (74%) followed by arterial (54%). GAPSS [29] and the prevalence of other vascular risk factors were similar between study groups.

### 2.2. Small Extracellular Vesicles (sEVs) in Patients with PAPS and aPL

The characterisation of sEVs revealed spherical structures with the characteristic cup-shaped morphology, and an average diameter of 174.2 ± 40.2 nm, 149.5 ± 48.2 nm and 172.8 ± 45.0 nm for PAPS, aPL patients and HD, respectively (Figure 1A). Western Blot analysis demonstrated the presence of sEV marker proteins, ALIX and TSG101 (Appendix A). Levels of circulating sEVs were significantly increased in patients with PAPS and aPL compared to HD (8.37 × 10^7^ PAPS, 8.38 × 10^7^ aPL vs. 1.19 × 10^7^ HD, *p* < 0.0001, Figure 1C). No significant differences in sEV quantity were observed by clinical manifestation or the type of antiphospholipid antibody (Appendix A). sEVS did not demonstrate procoagulant activity as measured by the STA-Procoag-PPL assay (Appendix A).

We compared sEV cell identity surface protein marker expression between study groups (*n* = 15, each group). We distinguished microvesicles from sEVs by size using scatter parameters of flow cytometer (Appendix A). sEVs from the three study groups predominantly exhibited a platelet marker CD41b (74.4%), and endothelial CD31 to a lesser extent (25.6%), similarly distributed amongst groups (Appendix A). Significantly increased of platelet-derived sEV surface CD62P marker was observed in PAPS and aPL patients compared with HD (*p* < 0.001, Figure 1D). Immunofluorescence images showed a differential uptake profile in HUVECs between disease groups (PAPS and aPL) and HD (92% and 90% vs. 12%, *p* < 0.001, Figure 1E). Additionally, increased monocyte sEV uptake occurred similarly for all study groups (Figure 1E). No significant uptake in platelets was observed.

### 2.3. sEVs from aPL and PAPS Patients Produce Endothelial Dysfunction

Endothelial dysfunction is a key factor for antiphospholipid syndrome pathogenesis. It has been associated with an increase in endothelial proliferation and the induction of inflammatory and procoagulant phenotypes [30]. We investigated whether aPL and PAPs sEVs could influence endothelial dysfunction by endothelial cell proliferation. After 24 h of sEVs incubation into HUVECs cells, we observed a significant increase in their proliferation compared to HD (*p* < 0.001, Figure 2A). Known cell proliferation pathways were analysed by RT-qPCR for RAS/MEK/ERK, transforming growth factor beta (TGF-β) and the phosphatidylinositol 3-kinase (PI3K)/AKT/mammalian target of rapamycin (mTOR) pathways. HUVECs exposed to sEVs from PAPS and aPL patients had increased gene expression levels of *PI3K1*, *AKT1* and *MTOR* compared to HD (fold change of 16.36 and 10.31, 4.48 and 3.10, 8.32 and 7.05, respectively, Figure 2B). We did not observe an effect on HUVEC apoptosis (Appendix A). No significant changes were observed in *TGFBR1*, *TGFB*, *SMAD2*, *SMAD3*, *MAPK1* and *MAPK3* and phosphorylated MEK/ERK pathway (Appendix A).

Next, we studied the effect of aPL and PAPs-derived sEVs in the induction of inflammatory and procoagulant molecules. sEVs induced an upregulation of cell adhesive molecules *VCAM1* and *ICAM1* in cultured HUVECs, both at protein level (*p* < 0.001, Figure 2C,D) and in gene expression (fold change of 6.57 and 5.85 for PAPS sEVs; fold change of 5.29 and 3.30 for aPL sEVs, Figure 2E). E-selectine expression was not significantly modified (Appendix A). Additionally, we observed a significant upregulation of transcriptor factor *NFKB1* (*p* < 0.005, Figure 2E), and its phosphorylated protein (Figure 2D). Changes in NF-ҡB related inflammatory cytokines (IL-1, TNF, IL-6, IL-8 or IFNα/γ, Appendix A) were not detected. Only sEVs from PAPS patients increased the expression of *F3* (TF) relative to the control group (2.98-fold change, Figure 2E).

### 2.4. PAPS sEVs Produced Monocyte Activation

Growing evidence supports the involvement of monocytes in APS pathogenesis. Mechanisms involved in their procoagulant and proinflammatory state following stimulation by aFLs include increased production of tissue factor and upregulation of NFκB, MEK-1/ERK and p38 MAPkinase pathways [31]. To study the molecular effect triggered by aPL and PAPS sEVs in monocytes, we incubated sEVs with human primary monocytes. Through mRNA gene expression analysis, we observed that only sEVs from PAPS differentially induced the expression of F3 (fold change of 5.60), transcriptor factor *NFKB1* (fold change of 5.80), the co-stimulatory factor *CD80* (fold change of 6.00) and inflammatory cytokines (*MCP1* and *IL12*) (*p* < 0.005, Figure 3A). Significant high levels of MCP-1 and IL-12 were detected in culture medium by ELISA assay (Figure 3B) and phosphorylated NF-ҡB by immunofluorescence (Figure 3C).

No changes were observed in p38 MAPK (*MAPK14*, *MAPK11*), MEK/ERK1 (*MAPK1*, *MAPK3*, *MA2PK1*), TLR-mediated signaling pathways (*TLR2/4*, *TRAF6*) and other relevant inflammatory cytokines (*TNFA*, *IFNA1*) (Appendix A). aPL- and HD-derived sEVs did not produce monocyte activation (Figure 3A–C).

### 2.5. Differential Expression Profiles of sEV-Derived miRNA between PAPS and aPL Patients

Since the biological effect of sEVs depends on their cargo and sEV-embedded miRNAs are relevant in gene expression regulation, we performed a miRNA-seq analysis of sEVs by study group (*n* = 8 each) to identify a miRNA profile. Patient characteristics are shown in Appendix A. Principal component analysis and hierarchical clustering distinguished nine differentially expressed (DE) miRNAs between aPL and HD and ten between PAPS vs. HD using the conditions of *p* adj value < 0.05 and fold change > 1.5. Comparing PAPS vs. aPL, no significant values were obtained with *p* adj value < 0.05, but we identified four differential miRNAs using *p* adj value < 0.15 (Figure 4A and Appendix A).

Only two common DE-derived miRNAs were identified between aPL vs. HD and PAPS vs HD groups (miR-483-3p and let-7a-3p) and one between PAPS vs. HD and PAPS vs. aPL (miR-326, Figure 4B). We also validated the top five miRNAs in each group by signal (marked bold in Appendix A). Only miR-326 and miR-483-3p were significantly DE. miR-326 was predominantly downregulated in sEVs from PAPS compared to aPL and HD (fold change decrease of 2.66 and 2.00, respectively, *p* < 0.001), whereas miR-483-3p was mainly overexpressed in PAPS and aPL groups compared to HD (fold change increase of 3.31 and 2.27, respectively) (Figure 4C).

### 2.6. sEV-Derived miRNAs Produce HUVEC Cell Activation via PTEN Pathway and Monocyte Activation via NOTCH1

We analysed whether the identified deregulated miRNAs could replicate the biological function observed by the study sEVs. First, we transfected mirVana miR-483-3p mimic into HUVECs to overexpress endogenous miR-483-3p (Appendix A). MiR-483-3p mimic transfected HUVECs had increased proliferation rates compared to control miR-mimic (fold change of 2.5, Figure 5A); upregulation of *PI3K1*, *AKT1* and *MTOR* (fold change increase of 12.3, 3.3 and 4.7, respectively, Figure 5B); *F3* and *EDN1* (fold change of 5.8, 2.0 and 3.5, respectively); *ICAM1* and *VCAM1* adhesion factors (fold change of 18.4 and 29.7, *p* < 0.001, Figure 5B) and *NFKB1* transcription factor (fold change of 8.1, Figure 6D). VCAM-1, ICAM-1 and increased phosphorylated NF-ҡB expression were confirmed at protein level (Figure 5C).

To identify the molecular mechanism by which miR-483-3p contributes to endothelial dysfunction, we retrieved validated target genes from three major miRNA-target datasets: miRecords, miRTarBase and miRWalk. We identified 10 candidate genes for miR-483-3p (Appendix A). Filtering by biological function, we identified *PTEN*, *IGF1*, *BBC3* and *CDK4* as potential targets of miR-483-3p in HUVECs. Of these, only *PTEN* transcript downregulation was confirmed by qRT-PCR analysis upon miR-483-3p overexpression in HUVECs cells (fold change decrease of 6.7, (Figure 5D and Appendix A). Transfection of miR-483-3p mimics resulted in suppression of the reporter luciferase activity containing regions of the 3′UTR of human PTEN (relative luciferase ratio FLuc/Rluc = 33% of reduction, *p* < 0.001) (Figure 5E,F). Next, we demonstrated that sEVs from PAPS and aPL incubated with HUVECs produced a *PTEN* downregulation (fold decrease of 2.2 and 1.5, respectively, Figure 5G).

Inhibition of miR-326 in monocytes induced upregulation of tissue factor gene (*F3*), the transcriptor factor *NFKB1*, the cytokines *MCP1* and *IL12* and co-stimulatory receptor *CD80* (fold increase of 2.7, 5.7, 3.1, 1.6 and 2.6, respectively, Figure 6A). MCP-1, IL-12 and phosphorylated NF-ҡB increased expression were also confirmed at protein level (Figure 6B,C). No effect was observed when miR-483-3p was upregulated (Appendix A).

Retrieval of validated genes as described above identified 14 candidates for miR-326 (Appendix A), of which *NOTCH1* was identified as a potential direct target using luciferase assay (Figure 6D,E). miR-326 inhibition or culture with PAPs sEVs induced an upregulation of *NOTCH1* in monocytes (fold increase of 6.2 and 7.9, respectively, *p* < 0.001, Figure 6F).

## 3. Discussion

In this study, we have demonstrated that patients with aPLs, with and without a history of thrombosis, have higher circulating sEVs than healthy donors. These sEVs were mainly of platelet origin and had endothelial and monocytes as target cells. We identified two novel sEV-derived miRNAs that could contribute to APS pathogenesis by promoting endothelial dysfunction, inflammation, and the production of procoagulant factors. Firstly, miR-483-3p was upregulated in patients with aPL antibodies and PAPS, with a predominant role in endothelial cells; we also identified PTEN, a negative regulator of the PI3K/Akt signaling pathway, as a target for miR-483-3p in HUVECS. Secondly, we found miR-326 to be downregulated uniquely in sEVs from PAPS patients with a pr\edominant role in monocyte activation and production of TF through NOTCH-1.

One study has characterised the presence of sEVs in APS [19]. The study detected similar quantities of sEVs originating from various hematopoietic cells in patients with APS and aPL-neg with idiopathic thrombosis. However, only sEVs from APS patients were specifically enriched in CD62P and CD133/1 surface expression, suggesting ongoing endothelial activation/damage and platelet activation in APS [19]. In line with that study, we have shown that patients with aPL antibodies also had an increased number of circulating sEVs compared with HD, mainly from an activated platelet origen. Large and medium EVs are frequently derived from platelets, make up approximately 70% of vesicles content in blood serum and have procoagulant activity [17,18]. We did not observe this in patient-derived sEVs procoagulant activity. But, we have shown that aPL and PAPS sEVs were highly engulfed by HUVEC cells and monocytes, causing endothelial cell dysfunction, and, only those sEVs from PAPS patients inducing monocyte activation. Platelet-derived sEVs containing miR-223, miR-339 and miR-31 can inhibit platelet-derived growth factor-β (PDGF-β) expression in vascular smooth muscle cells and contribute to atheromatosis by modifying the expression of adhesion molecules and TF [32]. Contrary to SLE and other connective tissue diseases, there are limited data on the direct role miRNA-derived sEVs in APS. In our study, for the first time, we performed a miRNA-seq in isolated from sEVs involved in APS cellular regulation and we identified two novel miRNAs, miR-483-3p and miR-326.

Endothelial dysfunction is a key pathological component of APS. Although in vivo analyses of endothelial dysfunction in APS have been reported, most research has been performed in vitro using models with endothelial cells exposed to either serum/plasma, monoclonal aPL, or IgGs isolated from APS patients [33,34,35]. These studies have described a reduction in endothelial cell nitric oxide synthesis, increase in endothelial proliferation [33], impairment of vascular remodeling and the induction of pro-inflammatory, pro-coagulant and pro-adhesive phenotypes [34,35]. In our study, we have shown that sEVs from PAPS- and aPL-carrier patients directly induce endothelial dysfunction in vitro. Previous studies have shown the role of miR-483 in endothelial cell dysfunction. On one hand, this plays a protective role by inhibiting the expression of TGF-β, CTGF, ACE1 and ET-1 [36]. On the other, it functions as a novel diagnostic biomarker in venous thromboembolism [37]. In our study, miR-483-3p overexpression increased endothelial cell proliferation and promoted an inflammatory-procoagulant state by producing proinflammatory cytokines, adhesion molecules (VCAM-1, ICAM-1) and TF via activation of the phosphatidylinositol 3-kinase–Akt (PI3K-Akt)-mammalian target of rapamycin complex (mTOR) and NF-ҡB pathway. Previous studies on APS have shown the relevance of this pathway. Oxidised oxLDL/β2GPI/anti-β2GPI antibody complex inhibited endothelial autophagy via PI3K/AKT/mTOR contributing to endothelial cell dysfunction [35]. Furthermore, mTORC2/Akt pathway mediates the promotion of platelet activation and induction of thrombosis by the anti-β2GP1 antibody [38]. In addition, by combining clinical studies with in vitro experiments, it has been shown that mTORC plays a role in the development of APS vasculopathy, with mTOR inhibitors able to reduce vascular proliferation in APS nephropathy [35,39]. In this study, we demonstrated that in HUVECs, miR-483 directly targets PTEN. Regulation of PTEN expression by miR-483-3p had been previously described in Wilms tumor cells [40]. sEV-derived miR-483-3p overexpression from PAPS and aPL patients may decrease PTEN gene expression, increasing PI3K/Akt signalling activity in HUVECs, a key link that stimulates endothelial proliferation via mTOR and inflammation via NF-ҡB signalling (Figure 7).

It is well known that monocytes play an active role in thrombogenesis, since, once activated, these cells express and release tissue factor (TF) [41]. aPL antibodies induce TF expression by activating, simultaneously and independently, the phosphorylation of MEK-1/ERK proteins, and the p38 MAP kinase-dependent nuclear translocation and activation of NF-ҡB/Rel proteins [42]. Aside from TF, aPL-activated monocytes contribute to the procoagulant state by triggering the production of proinflammatory cytokines, chemokines, and adhesion molecules which leads to the attraction of additional monocytes and T helper cells to the sites of inflammation [43,44]. In our study, we have demonstrated in vitro that interaction of PAPS sEVs with cultured monocytes increased the expression of TF, costimulatory CD80 molecules, NF-ҡB transcriptor factor and related inflammatory cytokines. Downregulation of miR-326 was only found in sEVs from PAPS patients. As with sEVs, downregulation of miR-326 in monocytes led to NF-ҡB signalling activation with increased production of TF, MCP-1 and IL-12. NOTCH1 is a described gene target of miR-326 [40] and induced inflammatory responses by activation of NF-ҡB pathway and increasing cytokine production (CCL2 and IL-12) in monocytes [45,46]. We demonstrated that miR-326 target NOTCH1 in monocyte. Consequently, sEV-derived miR-326 downregulation may activate monocytes by upregulating NOTCH signalling, promoting TF and cytokine production (Figure 7). miR-326 has been previously described as regulator of platelet apoptosis by targeting Bcl-xl, and antiapoptotic Bcl-2 family regulator [47]. However, we did not see this effect since our sEVS were not internalised by platelets.

In conclusion, our results demonstrated enhanced shedding and distinct biological properties of miRNA-derived sEVs between PAPS patients and aPL-carrier patients. These findings provide insights into their role in APS pathogenesis to be applied as disease biomarkers or as new targeted therapeutic approaches.

## 4. Materials and Methods

### 4.1. Patients

From 2017 to 2019, patients with primary APS (PAPS) (*n* = 50) according to International Consensus statement criteria [2] or with persistent aPLs without associated thrombotic or obstetric complications (aPL) (*n* = 30) were consecutively included from the outpatient rheumatology clinic at Vall d’Hebron Hospital. The control group consisted of HD (*n* = 30) without aPLs or any associated complications. The study was ethically approved by Vall d’Hebron Hospital Ethics Committee and all patients provided written informed consent prior to inclusion. Details of demographic characteristics, concomitant medications, comorbidities, and the Global Anti-Phospholipid Syndrome Score (GAPSS) [29] were obtained.

### 4.2. Sample Collection

Serum, plasma, and peripheral blood mononuclear cells (PBMCs) were obtained from each participant. All samples were processed within one hour of blood drawing. Monocytes were isolated from PBMCs from all participants, and platelets were obtained from HD whole blood. sEVs were isolated from plasma using the ExoQuick Plasma prep and Exosome precipitation kit (System Biosciences SBI, Palo Alto, CA, USA) and characterised by NanoSight, Cryo-TEM, western blot and flow cytometry analysis (LSR Fortessa, BD Biosciences, Erembodegem, Belgium, Appendix A). sEV quantification was determined via the FluoroCet Exosome Quantification Kit (System Biosciences SBI, Palo Alto, CA, USA) and sEV procoagulant activity by STA-Procoag-PPL (Stago, Asnieres, France). Details in Appendix A.

### 4.3. Anti-Phospholipid Antibody Detection

The detection of aCL and aβ2GPI in patient sera, and analysis of Lupus Anticoagulant (LA) was performed as described elsewhere [2,29] (details in Appendix A).

### 4.4. Small RNA-Sequencing of sEV-miRNAs

miRNAs contained in sEVs from aPL, PAPS and HD (*n* = 8) were extracted following the instructions of miRCURY RNA Isolation Kit—Cell & Plant (Exiqon, Woburn, MA, USA). miRNA samples were processed and sequenced (single end, 50 nts, 1 × 50, v4) by CNAG (Barcelona, Spain) on a HiSeq2500 Illumina device with a read depth of >10 M reads/sample. Subsequent quality control, data processing and analysis were performed as previously described [48,49,50]. Data were deposited in the Gene Expression Omnibus (NCBI) with the number GSE220791 (more details in Appendix A).

### 4.5. Cell Cultures

Primary umbilical human vascular endothelial cells (HUVECs, Innoprot, Derio, Spain), passages 2–5, were cultured in endothelial cell medium (ECM, Innoprot, Spain). Monocytes were cultured using RPMI-1640 medium (Gibco, Thermofisher Scientific, Carlsbad, CA, USA). Platelets were isolated and used within 2–3 h. Labelled sEVs were incubated with primary cells for 2, 4, 6, 8 and 24 h at 37 °C. sEV internalisation was analysed by fluorescence microscopy (Olympus BX61). Functional miRNA studies were performed in HUVECS and monocytes. Cells at ~70% confluence were transfected with mimic or anti-miR miRNA (mirVanaTM, Life technologies, Carlsbad, CA, USA) using Lipofectamine RNAiMAX Reagent (Life Technologies, Carlsbad, CA, USA).

### 4.6. RNA Extraction and RT-qPCR Gene Expression

HUVECS or monocytes cells were incubated with sEVs or miRNA transfected. After that, RNA extraction was performed using RNeasy Mini Kit following manufacturer’s instructions (Qiagen, Hilden, Germany). Once RNA was obtained, total RNA was reverse transcribed into cDNA using the High-Capacity cDNA Reverse Transcription Kit (Applied Biosystems). Gene expression was assessed by TaqMan gene expression assays (FAM dye labelled MGB probe, Applied Biosystems, Appendix A). Obtained data were normalised based on the expression of the endogenous control gene GAPDH (Hs02786624_g1) [51].

### 4.7. Apoptosis and Proliferation Assay

Cells were plated in 24-well plates and incubated with sEVs or transfected with corresponding anti/mimic miRNA. After 24 h, they were stained with Dead Cell Apoptosis Kit with Annexin V APC and SYTOX™ Green (Thermofisher, Carlsbad, CA, USA) and measured by flow cytometry. For proliferation assays, cells were plated in 96-well plates, incubated with sEVs or transfected. After 24 h, CyQUANT NF Cell Proliferation Assay Kit (Invitrogen, Carlsbad, CA, USA) was used following manufacturer’s instructions. Fold changes in fluorescence were calculated over cells incubated with sEVs from HD or over cells transfected with mimic or anti-control.

### 4.8. Luciferase Assay

Primary HUVECs or primary monocytes were co-transfected with the vector pEZX-MT01-PTEN 3′UTR or pEZX-MT01-NOTCH1 3′UTR and miR-483-3p or miR-326 mimics (10 μM) (mirVanaTM, Life technologies, Carlsbad, CA, USA) using DharmaFECT Duo transfect reagent (Thermo Fisher Scientific, Waltham, MA, USA) according to manufacturer protocols. After 24 h, the luciferase activity of firefly and Renilla was measured by Dual-Luciferase Reporter Assay System (Promega, Madison, WI, USA). Renilla luciferase was used as internal control.

### 4.9. Immunofluorescence on Primary Cells

After sEVs incubation or miRNA transfection, cells were washed with PBS and fixed for 15 min in 4% paraformaldehyde at 4 °C followed by permeabilization with 0.1% TritonX-100 for 10 min at room temperature. Blocking solution (BSA 5%) was added for 1 h at RT and primary antibodies were incubated overnight at 4 °C and secondary antibodies were added for 2 h at room temperature (Appendix A). DAPI was used to visualize the nucleus. Images were captured using Olympus BX61 microscope. Staining intensity was graded as 0 (no staining), 1 (weakly positive, <25% stained), 2 (moderately positive, 25–75% stained) or 3 (strongly positive, >75% stained).

### 4.10. Measurement of Cytokines Levels by Enzyme-Linked Immunosorbent Assay (ELISA)

Isolated primary monocytes from healthy donors were plated in 24-well plates and incubated with sEVs or miRNA-transfected. After 48 h, conditioned media were collected and used to measure cytokines levels. For quantify MCP-1 and IL-12 levels, we used “Human MCP-1 Elisa Kit” (873.030.096, Diaclone, Besançon, France) and “Human IL-12 p70 Elisa Kit” (ab100552, Abcam, Cambridge, UK) according to the respective manufacturer’s instructions.

### 4.11. Statistical Analysis

Statistical analysis was performed using SPSS 17.0 statistical software (SPSS, nc., Chicago, IL, USA) and graphs were generated using GraphPad Prism 7.0. Data are expressed as mean ± standard deviation. All experimental data were generated in triplicate, and experiments were repeated at least three times. *p* < 0.05 was considered to be statistically significant and statistical tests are indicated in the figures.

## Figures and Tables

**Figure 1 ijms-24-11607-f001:**
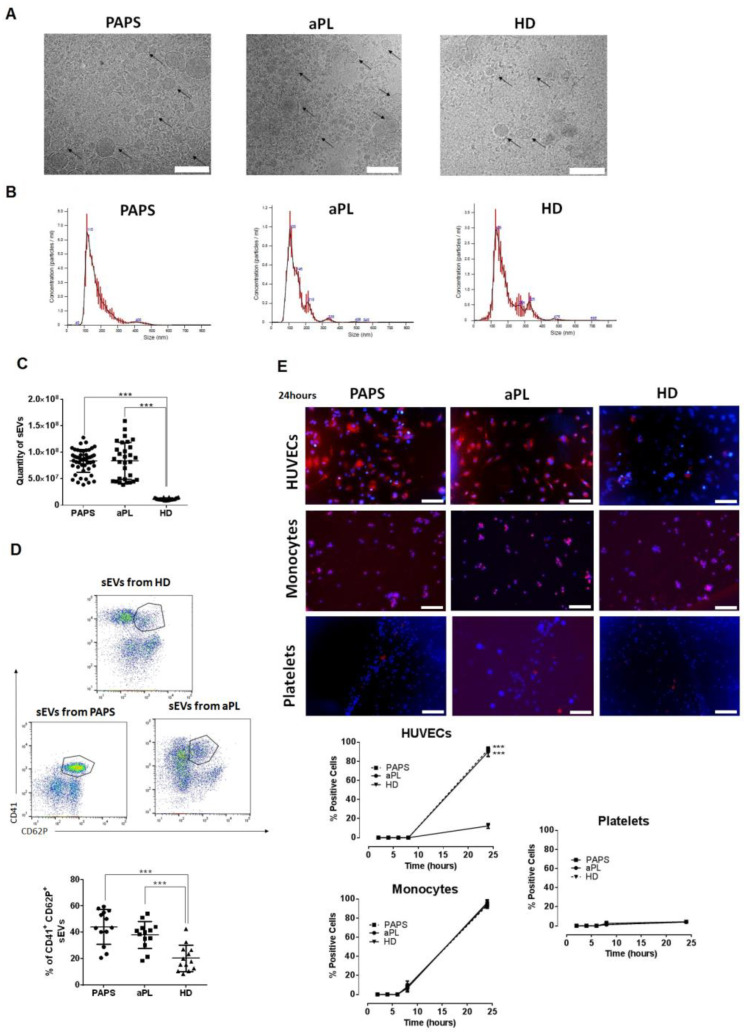
Isolation, characterization, origin and cellular uptake of sEVs from PAPS, aPL and HD samples. (**A**) Cryo-TEM images of sEVs isolated from plasma of patients with primary antiphospholipid syndrome (PAPS), patients with positive antiphospholipid antibodies (aPL) and healthy donors (HD). Small EVs have the typical round shape (black arrows). Scale bar = 500 nm. (**B**) Nanoparticle tracking analysis distribution of sEVs from plasma of PAPS, aPL patients and HD. The major concentration of vesicles from the three samples were around 100 nm in diameter. The experiment was repeated three times in total (error bars ± SEM, indicated in red). (**C**) Quantity of sEVs was calculated by fluorescence emission at 590–600 nm and using known standards calibrated samples. A significant increase in sEVs quantity was observed in plasma samples of PAPS patients and aPL patients in comparison with HD. *** *p* < 0.0001 by one-way ANOVA. (**D**) Surface protein CD62P evaluated by flow cytometry analysis showed that sEVs from PAPS and aPL are from activated platelets. *n* = 15 for each group. *** *p* < 0.001. (**E**) sEVs from PAPS, aPL and HD were stained and incubated with endothelial, platelet and monocytes cells at 2, 4, 6, 8 and 24 h to evaluate by immunofluorescence their cell internalization. Images were obtained after 24 h of incubation. Scale bar = 50 μm. Significant were obtained in comparison with HD group. *** *p* < 0.001.

**Figure 2 ijms-24-11607-f002:**
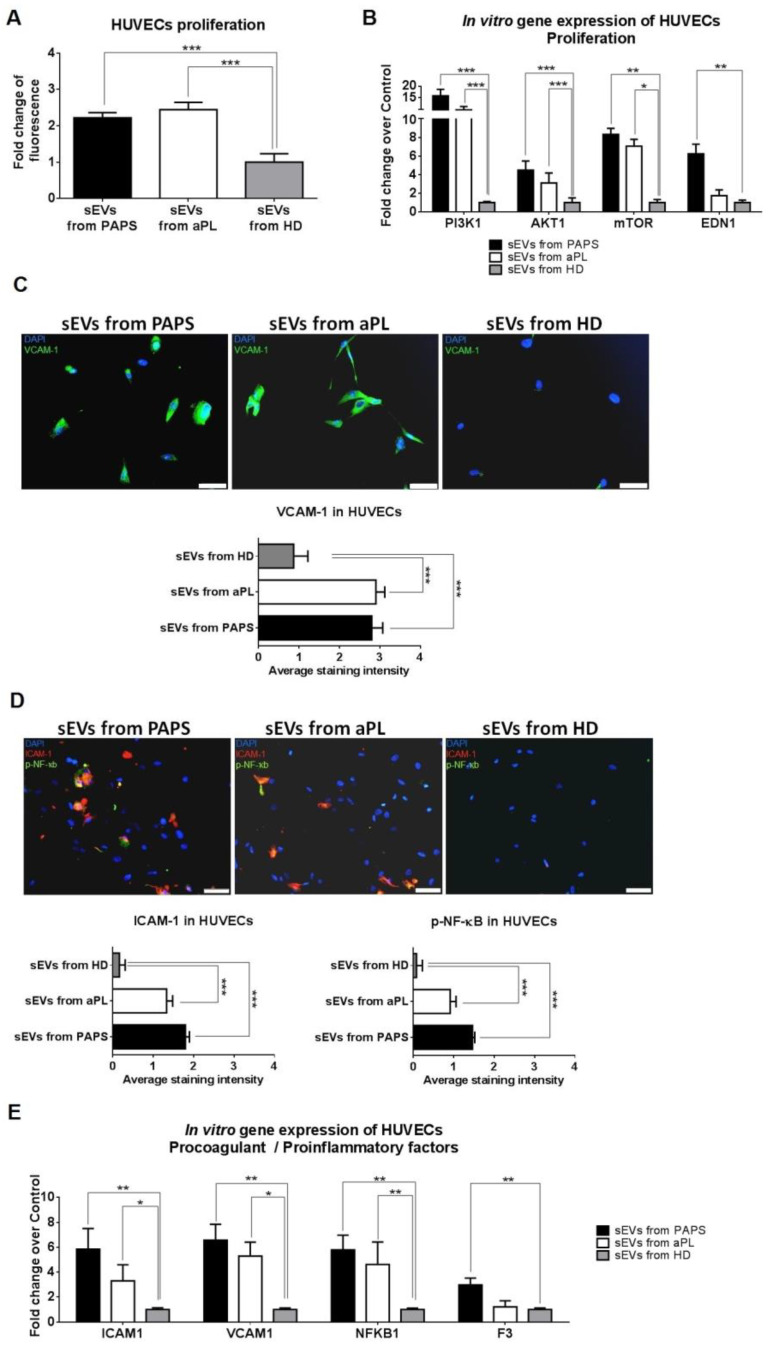
In vitro effect of sEVs from PAPS, aPL and healthy donors to HUVECs. (**A**) Proliferation assay was performed in HUVECs after incubation with sEVs from PAPS, aPL or HD (healthy donor). Fluorescence measurements were made using a microplate reader with excitation at 485 nm and emission detection at 530 nm and fold change was calculated over sEVs from HD. (**B**) Gene expression of proliferation in HUVECs after incubation with sEVs. Fold change was calculated over sEVs from HD. (**C**,**D**) Immunofluorescence in HUVECs after incubation with sEVs from PAPS or aPL. Green staining was used for VCAM-1 protein (**C**) and phosphorylated NF-ҡB (**D**). ICAM-1 was stained with red (**D**) and DAPI was used for nuclei cells (blue). Scale bar = 50 µm. In vitro experiments were performed a minimum of three replicates. 1-way ANOVA was performed to analyse difference between three groups. * *p* < 0.05, ** *p* < 0.005, *** *p* < 0.001. (**E**) Gene expression of procoagulant/proinflammatory factors in HUVECs after incubation with sEVs. Fold change was calculated over sEVs from HD. 1-way ANOVA was performed to analyse difference between three groups. * *p* < 0.05, ** *p* < 0.005.

**Figure 3 ijms-24-11607-f003:**
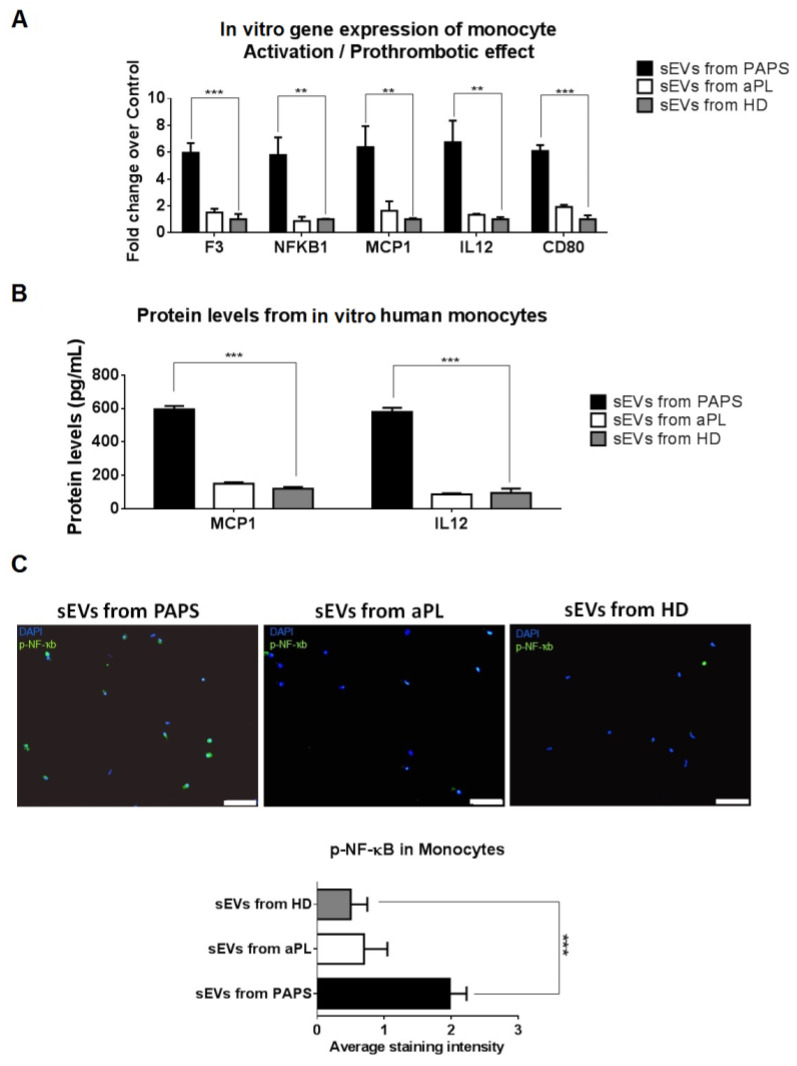
In vitro effect of sEVs from PAPS, aPL and healthy donors to monocytes. (**A**) Primary monocytes were incubated with sEVs for 24 h, and qPCR-RT was performed for quantify gene expression. (**B**) MCP-1 and IL-12 production (pg/mL) by monocytes obtained from healthy donors after 48 h of incubation of sEVs isolated from PAPS, aPL or HD (healthy donor). Values are means ± SD. 1-way ANOVA was performed to analyse difference between three groups. *** *p* < 0.001. (**C**) Immunofluorescence was performed in order to quantify phosphorylated NF-ҡB protein levels (green) in monocytes after incubation with sEVs. Nuclei cells are stained with DAPI (blue). Scale bar = 50 µm. ** *p* < 0.005, *** *p* < 0.001.

**Figure 4 ijms-24-11607-f004:**
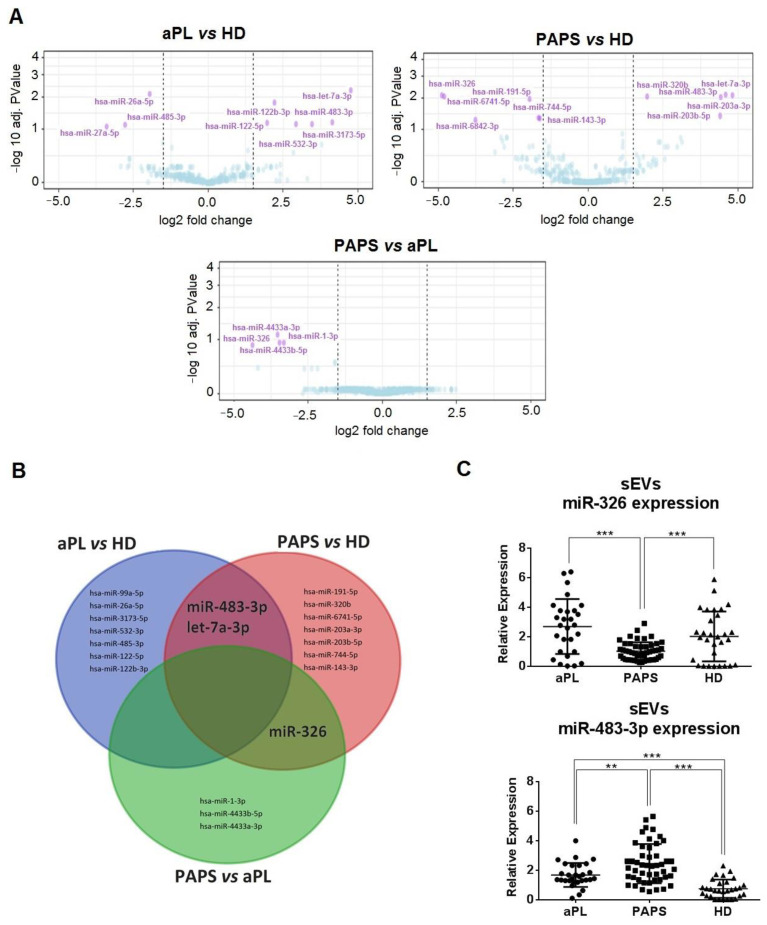
Differently expressed miR-326 and miR-483-3p in sEVs from PAPs and aPL patients. (**A**) Volcano plot of miRNA screening of sEV-derived miRNA expression for each comparison (aPL vs. HD, PAPS vs. HD and PAPS vs. aPL). Purple dots represent miRNAs that were significantly down or upregulated by log2 fold change of 1.5 (*p* value < 0.001 and *p* adj value < 0.15), paired *t*-test, *n* = 8 for each group. (**B**) Venn diagram showing overlapping miRNAs between each comparison (log2 fold change of 1.5, *p* value < 0.005 and *p* adj value < 0.15). (**C**) Verification of miRNA differences found in the miRNA-seq screening. The most significant miRNAs selected from screening were used for single assay qRT-PCR confirmation. Relative gene expression was calculated using the 2−^ΔΔCt^ method. Only miR-326 and miR-483-3p showed significant difference between sEVs from PAPS (*n* = 50), aPL (*n* = 30) and HD (*n* = 30). ** *p* < 0.005, *** *p* < 0.001, one-way ANOVA.

**Figure 5 ijms-24-11607-f005:**
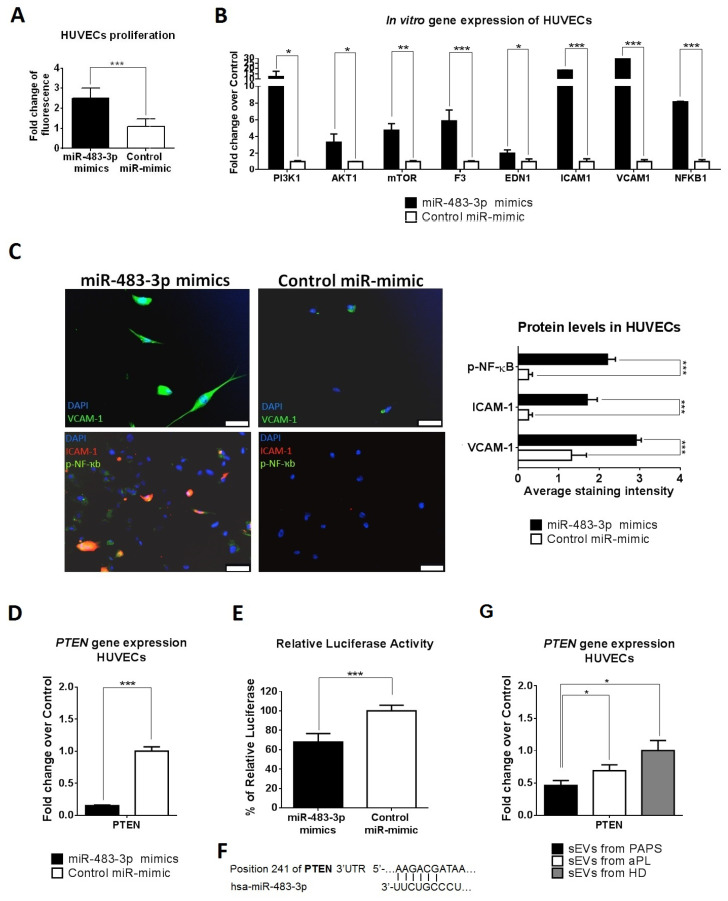
MiR-483-3p promotes endothelial proliferation/dysfunction by targets PTEN. (**A**) HUVECs were transfected with mimic miR-483-3p or mimic control. Proliferation assay showed high fluorescence signal in upregulated miR-483-3p HUVECs cells. Assay performed in triplicate. *t*-test, *** *p* < 0.001. (**B**) Gene expression of control miR-mimic and miR-483-3p mimics HUVECs cells were evaluated by qRT-PCR to study proliferation and coagulation/inflammatory/adhesion factors, *t*-test, * *p* < 0.05, ** *p* < 0.005 and *** *p* < 0.001. (**C**) Over miR-483-3p HUVECs has higher VCAM-1, phosphorylated NF-ҡB (green) and ICAM-1 (red) protein levels by immunofluorescence. DAPI was used to mark nucleic cells (blue). Scale bar = 50 µm. *t*-test, *** *p* < 0.001. (**D**) *PTEN* gene expression was evaluated in HUVECs with upregulation of miR-483-3p (miR-483-3p mimic). Significant upregulation was observed in comparison with control miR-mimic, *t*-test, *** *p* < 0.001. (**E**) Luciferase assay showed that miR-483-3p targets directly PTEN in HUVECs cells. Experiments were performed in triplicate. Results are expressed as the relative ratio of Firefly luciferase activity to Renilla luciferase activity using the negative control as reference. *t*-test, *** *p* < 0.001. (**F**) Putative binding site of miR-483-3p to *PTEN* gene was calculated by bioinformatics analysis (**G**) *PTEN* gene expression was evaluated in HUVECs cells after to be incubated with sEVs from HD, aPL or PAPS. *t*-test, * *p* < 0.05.

**Figure 6 ijms-24-11607-f006:**
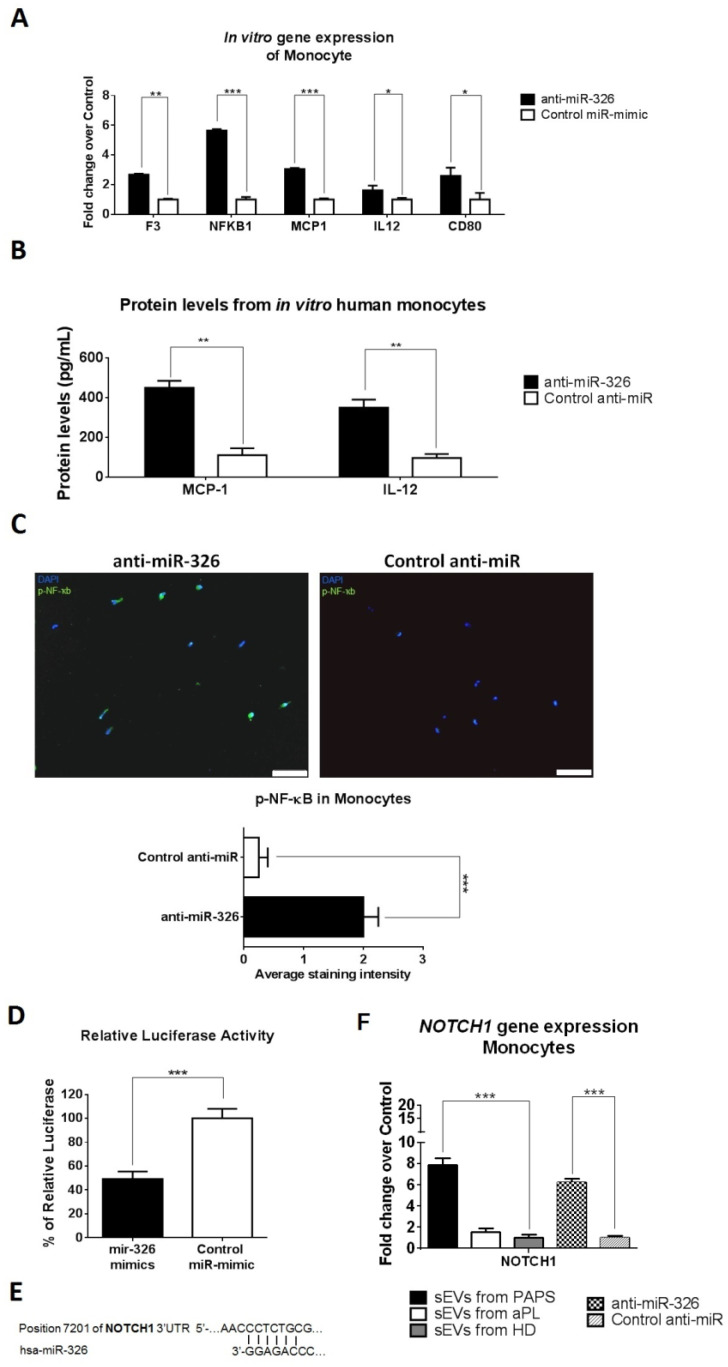
Inhibition of miR-326 in monocytes promotes NF-ҡB activation and related proteins via NOTCH pathway activation. (**A**) Monocytes were transfected with anti-miR-326 and qRT-PCR was performed to study gene expression. Experiments performed in triplicate, *t*-test, * *p* < 0.05, ** *p* < 0.005 and *** *p* < 0.001. (**B**) Protein levels of MCP-1 and IL-12 were measured in monocytes medium after transfection by ELISA assay. Values are means ± SD. *t*-test, ** *p* < 0.005. (**C**) Phosphorylated NF-ҡB protein levels were measured by immunofluorescence in monocytes cells (green staining). In vitro experiments were performed in triplicate. Scale bar = 50 µm. *** *p* < 0.001. (**D**) Luciferase assay showed that NOTCH1 is a gene target of miR-326 in human primary monocytes. In vitro experiments were performed in triplicate. Results are expressed as the relative ratio of Firefly luciferase activity to Renilla luciferase activity using the negative control as reference. *t*-test, *** *p* < 0.001. (**E**) Putative binding site of miR-326 to *NOTCH1* gene was calculated by bioinformatics analysis (**F**) *NOTCH1* was evaluated as gene targets of miR-326 in monocytes. Levels of gene expression were quantified by qRT-PCR and normalised using *GADPH* as endogenous control. *t*-test was performed between anti-miR-326 vs. control anti-miR and one-way ANOVA between sEVs from HD and from aPL or PAPS. *** *p* < 0.001.

**Figure 7 ijms-24-11607-f007:**
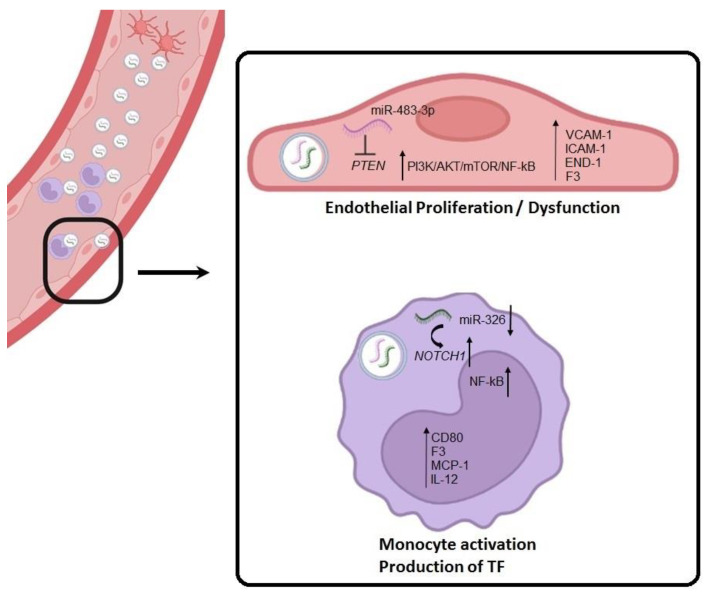
Proposed mechanism for the role of sEVs in APS pathogenesis. Proposed mechanism for sEVs-derived miRNA in APS pathogenesis. Activated platelet release sEVs that are going into endothelial and monocyte cells. Inside sEVs there is an upregulation of miR-483-3p and a downregulation of miR-326. In endothelial cells, miR-483-3p targets PTEN inducing an upregulation of PI3K-AKT-mTOR and NF-ҡB pathway producing endothelial proliferation and dysfunction with high levels of VCAM-1, ICAM-1, END-1 and tissue factor. Low levels of miR-326 provoke a monocyte activation by the upregulation of its target gene (NOTCH1) and co-stimulatory factors (CD80). Inflammatory response regulated also by NF-ҡB pathway increase in cytokines production (MCP-1 and IL-12) and procoagulant factors (tissue factor).

**Table 1 ijms-24-11607-t001:** Characteristics of the study patients at baseline.

Characteristic	PAPS (*n* = 50)	aPL (*n* = 30)	HD (*n* = 30)	*p* Value ^‡^
**Female sex, n (%)**	30 (60.0)	20 (66.6)	20 (66.6)	0.637
**Median age (IQR), y**	51 (44–60)	54 (44–60)	49 (40–59)	0.198
**Median duration of APS (IQR), y**	9.4 (3–14)	9.9 (4–12)	na	0.597
**PAPS clinical criteria for initial anticoagulation, n (%)**				
Venous thrombosis	37 (74.0)	0 (0)	0 (0)	<0.001
Arterial thrombosis	25 (50.0)	0 (0)	0 (0)	<0.001
Both arterial and venous	12 (24.0)	0 (0)	0 (0)	<0.001
**Obstetrical medical history, n (%)**	11 (22.0)	0 (0)	0 (0)	<0.001
**Other non-thrombotic manifestations, n (%)**				
Valvular heart disease *	19 (38.0)	2 (6.7)	0 (0)	0.002
Renal thrombotic microangiopathy	9 (18.0)	1 (3.3)	0 (0)	0.081
Livedo reticularis	23 (46.0)	1 (3.3)	0 (0)	<0.001
Migraine	23 (46.0)	4 (13.3)	0 (0)	0.003
Thrombocytopenia	13 (26.0)	5 (16.7)	0 (0)	0.414
**Laboratory profile at inclusion, n (%)**				
Lupus anticoagulant	50 (100)	30 (100)	nd	1.000
Lupus anticoagulant alone	7 (14.0)	3 (10.0)	nd	0.736
IgG/IgM antibodies				
aCL	44 (88.0)	24 (80.0)	nd	0.351
Anti-β_2_GPI	41 (82.0)	23 (76.7)	nd	0.577
Anti-aPS/PT	15 (30.0)	10 (33.3)	nd	0.806
IgG aCL, GPL units	153.5 (152.8)	141.0 (162.7)	nd	0.671
IgM aCL, MPL units	26.9 (53.3)	16.5 (20.3)	nd	0.309
IgG anti-β_2_GPI, GPL units	162.3 (217.4)	190.15 (241.0)	nd	0.596
IgM anti-β_2_GPI, MPL units	37.5 (73.3)	25.5 (47.6)	nd	0.426
**Lupus anticoagulant and IgG aCL and IgG anti-β_2_GPI antibodies, n (%)**	41 (82%)	25 (83.3)	nd	1.000
**GAPSS risk for thrombosis ^†^**				
Mean score (SD)	15.1 (2.1)	14.2 (2.7)	na	0.133
Score, n (%)				
<10	1 (2.0)	1 (3.3)	na	1.000
10 to <15	32 (64.0)	13 (43.3)	na	0.103
≥15	17 (34.0)	16 (53.3)	na	0.105
**Aspirin use, n (%)**	10 (20.0)	25 (83.3)	0 (0)	<0.001
**Antimalarial or immunosuppressive therapy**	0 (0)	0 (0)	0 (0)	1.000
**Vitamin K antagonist therapy, n (%)**	50 (100)	0 (0)	0 (0)	<0.001
**Coexisting cardiovascular risk factors, n (%)**				
Smoking	20 (40.0)	10 (33.3)	12 (40.0)	0.637
Dyslipedemia	23 (46.0)	11 (36.7)	5 (16.7)	0.487
Diabetes mellitus	8 (16.0)	3 (10.0)	0 (0)	0.522
Hypertension	21 (42.0)	10 (33.3)	0 (0)	0.485

PAPS = Primary antiphospholipid syndrome. aPL: Patients with persistent antiphospholipid antibodies without thrombotic or obstetric complication; HD = healthy donor; aPS/PT = anti-phosphatidylserine/prothrombin antibodies; na = not applicable; nd = not determined; *n* = Number; aCL = anticardiolipin; IgG = Immunoglobulin G; IgM = Immunoglobulin M; β_2_GPI = Beta-2-glycopotein I; SD = Standard deviation; GAPSS = Global Anti-Phospholipid Syndrome Score; GPL = IgG phospholipid; IQR = interquartile range; MPL = IgM phospholipid. Disease duration: time from the diagnosis of the condition (PAPS or aPL) to the study sample. * Cardiac valvular disease was defined by an echocardiographic detection of lesions and/or regurgitation and/or stenosis of mitral and/or aortic valve according to the actual definition of APS-associated cardiac valve disease [3]. ^†^ GAPSS [27]. This is a categorical score derived from the combination of independent risk for thrombosis and pregnancy loss (PL), considering the aPL profile, conventional cardiovascular risk factors and the autoimmune antibody profile that was developed and validated in a cohort of SLE patients. The GAPSS score ranges from 1 to 20, with higher scores indicating an increased risk. ^‡^ *p* value was calculated between PAPS and aPL groups, fisher exact test for categorial variables and *t*-test for continuous variables.

## Data Availability

Data were deposited in the Gene Expression Omnibus (NCBI) with the number GSE220791 at https://www.ncbi.nlm.nih.gov/geo/query/acc.cgi?acc=GSE220791 (accessed on 5 July 2023).

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
