# Peer review of "Small-Extracellular-Vesicle-Derived miRNA Profile Identifies miR-483-3p and miR-326 as Regulators in the Pathogenesis of Antiphospholipid Syndrome (APS)"

_ijms, 2023, doi:10.3390/ijms241411607_

Round 1
Reviewer 1 Report
The paper describes characterisation of EV from patients with APS including upregulation of mir483p and down regulation of mir326. They then go on to show that these mir are able to regulate some of the characteristic gene expression seen in endothelial cells and monocytes from these patients.
I am not convinced these are sEV / exosomes since you have characterised surface proteins more usually seen on the surface of "microvesicles" . I think it would be better to simply refer to these as EV, to avoid misinterpretation. Show some representative plots of the FCM data.
Fig5 below panel E - Fig6 below panel D diagrams showing the PTEN 3'-UTR and NOTCH1 3'-UTR need to be separately referred to in the legend
I would recommend a glossary with the abbreviations - there are many and it was hard to follow at times
some minor grammatical errors throughout, which need to be addressed
there are numerous minor grammatical and typographical errors which need to be addressed
Author Response
The paper describes characterisation of EV from patients with APS including upregulation of mir483p and down regulation of mir326. They then go on to show that these mir are able to regulate some of the characteristic gene expression seen in endothelial cells and monocytes from these patients.
- I am not convinced these are sEV / exosomes since you have characterised surface proteins more usually seen on the surface of "microvesicles" . I think it would be better to simply refer to these as EV, to avoid misinterpretation. Show some representative plots of the FCM data.
We followed the definition of EVs following the “Minimal information for Studies of Extracellular Vesicles guidelines (”MISEV2018”) for The International Society for Extracellular Vesicles (ISEV). According to this report, EV subtypes are classified by “physical characteristics” of EVs such as size (“small EVs” (sEVs) < 100nm or < 200nm, and “medium/large EVs” (m/lEVs) > 200nm), or density (low, middle, high, with each range defined)”. Based on their definition, our extracellular vesicles should be classified as small extracellular vesicles because they have a mean size between 100-200 nm: 174.2±40.2 nm, 149.5±48.2 nm and 172.8±45.0 nm for PAPS, aPL patients and HD (Page 2, lines 24-25).
In addition, as in this study analysing small extracellular vesicles in antiphospholipid samples (Štok, U. et al. Characterization of Plasma-Derived Small Extracellular Vesicles Indicates Ongoing Endothelial and Platelet Activation in Patients with Thrombotic Antiphospholipid Syndrome. Cells. 2020; 9, 1211), we used flow cytometry to distinguish microvesicles from sEVs. We followed the methodology of “Fortuanto D, et al. Extracellular vesicles isolated from porcine seminal plasma exhibit different tetraspanin expression profile. Scientific Reports. 2019; 9: 11584” to distinguish extracellular vesicles from microvesicles.
We have included the representative plots of flow cytometry analysis in the supplementary data (Figure S4 and Figure S5) and we have included this annotation in the text in order to avoid misinterpretation: “We distinguished microvesicles from sEVs by size using flow cytometer scatter parameters (Figure S4).” (Page 2, lines 33-34).
Regarding the presence of proteins on the surface of sEVs, several reports have shown that sEV outer membrane is a phospholipid bilayer containing many different membrane proteins. Some specific membrane proteins on the surface of sEV have been used as markers for isolation and identification of the sEV. The membrane proteins of sEV include integral membrane proteins, peripheral membrane proteins and lipid anchoring membrane proteins. Many of the membrane proteins are components of ESCRT complexes that are incorporated into sEVs during their formation. Others are originated from the plasma membranes and involved in cell-to-cell communication, and hence responsible for producing cellular effects in the targeted cells. Common membrane proteins of sEVs include Tetraspanin proteins (CD9, CD63, CD81), PD-L1, Integrins; Wnt protein, ALIX, Syntenin, HSPs, tenascin C; GPC1, Rabs, Flotillin (Huang D et al. Advances in Biological Function and Clinical Application of Small Extracellular Vesicle Membrane Proteins. Front Oncol 2021; 20, 11:675940).
- Fig5 below panel E - Fig6 below panel D diagrams showing the PTEN 3'-UTR and NOTCH1 3'-UTR need to be separately referred to in the legend
We have separated the figures following your suggestion.
- I would recommend a glossary with the abbreviations - there are many and it was hard to follow at times
We have included a glossary with all the abbreviations (Page 1).
- some minor grammatical errors throughout, which need to be addressed
We have revised all manuscript to correct the grammatical errors.

Reviewer 2 Report
In the article entitled “Small extracellular vesicles derived miRNA profile identifies miR-483-3p and miR-326 as regulators in the pathogenesis of primary antiphospholipid syndrome (PAPS)”, provide evidence that miRNA-derived sEVs contribute to APS pathogenesis by producing endothelial cell proliferation, monocyte activation and adhesion/procoagulant factors.
Presented data in this paper are important for antiphospholipid syndrome focused groups. However, several issues must be addressed before considering the manuscript for publication.
Comment 1: Page 1 line 48: "but, to date, only one study has evaluated the role of small EVs (sEVs) in APS pathogenesis [20]." Avoid using absolute terms.
Comment 2: Page 4 line 120, figure 1d: According to the article, the samples come from three groups, but the differences among these groups are not shown.
Comment 3: Page 5/7 line 127/163: Please elaborate briefly on the significance of cell proliferation and inflammatory factor expression detection in these parts (2.3/2.4).
Comment 4: Page 5 line 132: "analysed by RT-qPCR for RAS/MEK/ERK", If you want to detect the signal pathway, protein expression levels should be detected, especially the modification levels, such as changes in phosphorylation levels of MEK/ERK.
Comment 5: Page 5 line 137: There are a lot of abbreviations in the manuscript, please define them the first time they appear and please write the correct gene or protein name, such as the incorrectly written--INF.
Comment 6: Page 6, figure 2b: The numbers on the Y-axis are not clear, please revise the figure.
Comment 7: Page 8, line 195: Please delete the vertical line.
Comment 8: Page 8, line 211: Why not detect the expression level of let-7a-3p, and according to Table S2. hsa-miR-483-3p in the aCL vs HD comparison, p adj value is more than 0.05, so please correct figure 4b accordingly.
Comment 9: Page 8 and 9: Please briefly elaborate on the latest research on miR-483 and 326, especially in this disease model.
Comment 10: Page 9, line240: Why not detect the internal reference Renilla luciferase in this part of the experiment?
Comment 11: Page 10, figure 5E: In verifying the direct relationship between miRNA and its target, detection should also be performed after the target site mutation.
Comment 12: Page 17, line 501: Delete line 501. At the same time, the references in the manuscript are too old, with no papers from the last two years.
Comment 13: Materials and Methods: Please be consistent and add or remove cities for all compounds/media/reagents mentioned in the manuscript. Please use the same format for all reagents.

Author Response
In the article entitled “Small extracellular vesicles derived miRNA profile identifies miR-483-3p and miR-326 as regulators in the pathogenesis of primary antiphospholipid syndrome (PAPS)”, provide evidence that miRNA-derived sEVs contribute to APS pathogenesis by producing endothelial cell proliferation, monocyte activation and adhesion/procoagulant factors.
Presented data in this paper are important for antiphospholipid syndrome focused groups. However, several issues must be addressed before considering the manuscript for publication.
Comment 1: Page 1 line 48: "but, to date, only one study has evaluated the role of small EVs (sEVs) in APS pathogenesis [20]." Avoid using absolute terms.
We have rephrased it.
Comment 2: Page 4 line 120, figure 1d: According to the article, the samples come from three groups, but the differences among these groups are not shown.
Characteristics of the two groups of patients (PAPS and aPL) are shown in Table 1. We have also added the characteristics of the healthy donor group in the same Table 1. We explain these characteristics in the main text (Clinical characteristics, Page 2, lines 21-27).
Percentages of sEVs by source origin in the three study groups were similar. We have rephrased the figure to clarify the results:
We have also rephrased the sentence in the manuscript (Page 2, lines 39-41) and included the figure in supporting information (Figure S4).
Comment 3: Page 5/7 line 127/163: Please elaborate briefly on the significance of cell proliferation and inflammatory factor expression detection in these parts (2.3/2.4).
We have added an explanation about the significance of cell proliferation and inflammatory factor in endothelial dysfunction and monocyte activation (Parts 2.3 and 2.4, Page 5 lines 1-4, Page 6 lines 7-8 and Page 8 lines 1-3).
Comment 4: Page 5 line 132: "analysed by RT-qPCR for RAS/MEK/ERK", If you want to detect the signal pathway, protein expression levels should be detected, especially the modification levels, such as changes in phosphorylation levels of MEK/ERK.
Following your suggestion, we repeated the in vitro incubation of aPL, PAPs and healthy donors sEVs in HUVECs to extract protein lysates and evaluate phosphorylation levels of MEK/ERK by western blot. We performed it in triplicate. In previous in vitro experiments, we did not extract protein lysate since we used all samples for RNA extraction.
We used tubulin-α as control (Proteintech, mouse anti-human, 16801256) and p-ERK 1/2 (Thr202/Tyr204) antibody (Elabscience, rabbit anti-human, E-AB-70310) to evaluate phosphorylation protein. We did not observe a significant increase of p-ERK 1/2 in HUVECs after incubation of aPL or PAPS sEVs in comparison with HD sEVs:
These results had been included in the text (Page 5, line 14, Figure S7)
Comment 5: Page 5 line 137: There are a lot of abbreviations in the manuscript, please define them the first time they appear and please write the correct gene or protein name, such as the incorrectly written--INF.
We have included a glossary with all the abbreviations (Page 1). They had been defined in the text and we have corrected gene/protein names following your suggestion.
Comment 6: Page 6, figure 2b: The numbers on the Y-axis are not clear, please revise the figure.
We have corrected it.
Comment 7: Page 8, line 195: Please delete the vertical line.
We have deleted it
Comment 8: Page 8, line 211: Why not detect the expression level of let-7a-3p, and according to Table S2. hsa-miR-483-3p in the aCL vs HD comparison, p adj value is more than 0.05, so please correct figure 4b accordingly.
We evaluated let-7a-3p expression levels in our patient cohort s (aPL, PAPS and HD, N=30, 50 and 30, respectively) but we did not detect differences between them:
We corrected figure 4B legend and we included p adj value less than 0.15, in order to be consistent with Table S2.
Comment 9: Page 8 and 9: Please briefly elaborate on the latest research on miR-483 and 326, especially in this disease model.
Deep research of the literature was done using Pubmed, Igenuity Pathway and ChatGPT using as key words “miR-326” or “miR-483” and “antiphospholipid syndrome”. We obtained 0 results. We have changed the key words to “microRNA-326” or “microRNA-483” but again we obtained 0 results.
To update miRNA information in antiphospholipid syndrome, we performed a search using “miRNA” or “microRNA” and “antiphospholipid syndrome”, in this case we obtained 13 and 17 results, respectively. From them we have included these two reviews to increase the quality of our manuscript (references 25 and 26):
1) Pérez-Sánchez, L; Patiño-Trives, AM.; Aguirre-Zamorano, MA.; Luque-Tévar, M.; Ábalos-Aguilera, MC.; Arias-de la Rosa, I.; Seguí, P.; Velasco-Gimena, F.; Barbarroja, N.; Escudero-Contreras, A. et al. Characterization of antiphospholipid syndrome atherothrombotic risk by unsupervised integrated transcriptomic analyses. Arterioscler Thromb Vasc Biol. 2021; 41, 865-877.
2) Lopez-Pedrera, C; Barbarroja, N.; Patiño-Trives, AM.; Collantes, E.; Aguirre, MA.; Perez-Sanchez, C. New biomarkers for atherothrombosis in antiphospholipid syndrome: genomics and epigenetics approaches. Front Immunol. 2019; 10, 764.
In addition, we performed the search using “miR-483” or “miR-326” and “thrombosis”. In this case, we only found two papers that we have also included in our manuscript (references 37 and 47):
1) Xiang, Q.; Zhang, H-X.; Wang, Z.; Liu, Z-Y.; Xie, Q-F.; Hu, K.; Zhang, Z.; Mu, G-Y.; Ma, L-Y. et al. The predictive value of circulating microRNAs for venous thromboembolism diagnosis: A systematic review and diagnostic meta-analysis. Thromb Res. 2019; 181, 127-134
2) Yu, S.; Huang, H.; Deng, G.; Xie, Z.; Ye, Y.; Guo, R.; Cai, X.; Hong, J.; Qian, D.; Zhou, X. et al. miR-326 targets antiapoptotic Bcl-xL and mediates apoptosis in human platelets. PLoS One. 2015; 10: e0122784.
Comment 10: Page 9, line240: Why not detect the internal reference Renilla luciferase in this part of the experiment?
We used Dual-Luciferase Reporter assay system that detects firefly and Renilla luciferase activity in a single sample. Firstly, we transfected our cells with a vector pEZX-MT06 from GeneCopoeia that has reporter gene of firefly luciferase, tracking gene Renilla luciferase and miRNA target gene. At the same time, we also transfected the cells with the miRNA, in our case, miR-483-3p or miR-326. The structure of the vector included the region of miRNAs target gene as shown in the figure:
Later, we detected luciferase activity by Luc-Pair Luciferase Assay kit. We detected both luciferase activity, firefly and Renilla, sequentially. Renilla luciferase was used as internal control.
We have added in the manuscript (Page 10, line 29 and Figure 5E and 6D) and in Material and Methods (Page 19, line 28-30).
Comment 11: Page 10, figure 5E: In verifying the direct relationship between miRNA and its target, detection should also be performed after the target site mutation.
We observed that target genes (PTEN and NOTCH1) are downregulated when the levels of miR-483-3p or miR-326 are increased, respectively. We observed this downregulation after cell incubation with sEVs from patients loaded with miR-483-3p or miR-326 and also after cell transfection of the corresponding mimic or anti-miRNA. In consequence, we demonstrated that there is a relation between PTEN or NOTCH1 downregulation levels with the high levels of miRNA. However, we wanted to be sure so we performed a luciferase assay. Using bioinformatic tools we chose a putative target site. We observed a decrease of luciferase activity due to miRNA binding in the target site. This methodology has been described in these manuscripts:
- D’Onofrio, N., et al. (2023). MiR-27b attenuates mitochondrial oxidative stress and inflammation in endothelial cells. Redox Biol doi: 10.1016/j.redox.2023.102681
- Sun, B., et al. (2023). Engineered induced-pluripotent stem cell derived monocyte extracellular vesicles alter inflammation in HIV humanized mice. Extracell Vesicles Circ Nucl Acids doi: 10.20517/evcna.2022.1
- Mongiorgi, S., et al. (2023). A miRNA screening identifies miR-192-5p as associated with response to azacitidine and lenalidomide therapy in myelodysplastic syndromes. Clin Epigenetics doi: 10.1186/s13148-023-01441-9
In the cited manuscripts and in our manuscript, mutations in target site were not performed because the focus was not to study exactly the functionality of miRNA target site or the importance of each nucleotide in the miRNA target site. The identification and characterization of miRNA targets is thus a fundamental problem in biology. miRNAs regulate target genes by binding to 3′ untranslated regions (3′UTRs) of target mRNAs, and multiple binding sites for the same miRNA in 3′UTRs can strongly enhance the degree of regulation (Fang et al. The impact of miRNA target sites in coding sequences and in 3’UTRs. PLoS ONE. 2011; 6:E18067). miRNA could be targeting the gene in multiple sites. Our manuscript is not focused in the knowledge of which nucleotide of miRNA. For this reason, and like the other cited manuscript methodology, we have decided not to include the proposed experiment in the manuscript.
Comment 12: Page 17, line 501: Delete line 501. At the same time, the references in the manuscript are too old, with no papers from the last two years.
We have deleted the line and we have included papers that are more recent (references 25, 26, 36 and 38):
1) Pérez-Sánchez, L; Patiño-Trives, AM.; Aguirre-Zamorano, MA.; Luque-Tévar, M.; Ábalos-Aguilera, MC.; Arias-de la Rosa, I.; Seguí, P.; Velasco-Gimena, F.; Barbarroja, N.; Escudero-Contreras, A. et al. Characterization of antiphospholipid syndrome atherothrombotic risk by unsupervised integrated transcriptomic analyses. Arterioscler Thromb Vasc Biol. 2021; 41, 865-877.
2) Lopez-Pedrera, C; Barbarroja, N.; Patiño-Trives, AM.; Collantes, E.; Aguirre, MA.; Perez-Sanchez, C. New biomarkers for atherothrombosis in antiphospholipid syndrome: genomics and epigenetics approaches. Front Immunol. 2019; 10, 764.
3) Shang, F.; Guo, X.; Chen, Y.; Wang, C.; Gao, J.; Wen, E.; Lai, B.; Bai, L. Endothelial microRNA-483-3p is hypertension-protective. Oxid Med Cell Longev. 2022; 2022, 3698219.
4) Tang, Z.; Shi, H.; Chen, C.; Teng, J.; Dai, J.; Ouyang, X.; Liu, H.; Hu, Q.; Cheng, X.; Ye, J. et al. Activation of platelet mTORC2/Akt Pathway by anti-β2GP1 antibody promotes thrombosis in antiphospholipid syndrome. Arterioscler Thromb Vasc Biol. 2023. Online ahead of print.
Comment 13: Materials and Methods: Please be consistent and add or remove cities for all compounds/media/reagents mentioned in the manuscript. Please use the same format for all reagents.
We have corrected it following your suggestion.
